# Prediction of the Formation of Reactive Metabolites by A Novel Classifier Approach Based on Enrichment Factor Optimization (EFO) as Implemented in the VEGA Program

**DOI:** 10.3390/molecules23112955

**Published:** 2018-11-13

**Authors:** Angelica Mazzolari, Giulio Vistoli, Bernard Testa, Alessandro Pedretti

**Affiliations:** 1Dipartimento di Scienze Farmaceutiche, Università degli Studi di Milano, Via Mangiagalli, 25, I-20133 Milano, Italy; angelica.mazzolari@unimi.it (A.M.); giulio.vistoli@unimi.it (G.V.); 2University of Lausanne, CH-1015 Lausanne, Switzerland; b.testa2908@bluewin.ch

**Keywords:** reactive metabolite, toxicity prediction, machine learning, enrichment factor, unbalanced datasets

## Abstract

The study is aimed at developing linear classifiers to predict the capacity of a given substrate to yield reactive metabolites. While most of the hitherto reported predictive models are based on the occurrence of known structural alerts (e.g., the presence of toxophoric groups), the present study is focused on the generation of predictive models involving linear combinations of physicochemical and stereo-electronic descriptors. The development of these models is carried out by using a novel classification approach based on enrichment factor optimization (EFO) as implemented in the VEGA suite of programs. The study took advantage of metabolic data as collected by manually curated analysis of the primary literature and published in the years 2004–2009. The learning set included 977 substrates among which 138 compounds yielded reactive first-generation metabolites, plus 212 substrates generating reactive metabolites in all generations (i.e., metabolic steps). The results emphasized the possibility of developing satisfactory predictive models especially when focusing on the first-generation reactive metabolites. The extensive comparison of the classifier approach presented here using a set of well-known algorithms implemented in Weka 3.8 revealed that the proposed EFO method compares with the best available approaches and offers two relevant benefits since it involves a limited number of descriptors and provides a score-based probability thus allowing a critical evaluation of the obtained results. The last analyses on non-cheminformatics UCI datasets emphasize the general applicability of the EFO approach, which conveniently performs using both balanced and unbalanced datasets.

## 1. Introduction

The capacity of drugs and other xenobiotics to generate electrophilic reactive metabolites (RMs) is an unwanted property that should be carefully avoided during the design and development of drug candidates [1,2]. This is easily explained by considering that RMs can couple with nucleophilic sites within endogenous molecules forming stable covalent adducts endowed with clear toxic effects [1]. Even though detailed molecular mechanisms of toxication are not always well understood, and probably RMs are not the only trigger factor, a direct link between their formation and idiosyncratic adverse drug reactions (IADRs) is widely accepted [1]. Moreover, RMs are also involved in drug-induced liver injury (DILI) along with other factors such as a marked lipophilicity and high daily drug doses [2]. Not to mention that, when covalent modifications target DNA, RMs generation can result in mutagenicity [3].

Even though endogenous protective mechanisms (many of them based on the marked scavenging effects of glutathione (GSH), are able to detoxify these reactive species, the generation of RMs should be minimized or, better, essentially avoided in new drug candidates, especially considering that GSH levels are often markedly lowered under several pathological conditions such as oxidative-based diseases [4].

Several in silico approaches have been proposed with a view to minimizing the risk of RM generation [1]. To predict the metabolite’s reactivity, two different scenarios can be figured out. Firstly, and when all major metabolites a given drug candidate can generate have been characterized, several computational approaches mainly based on stereo-electronic descriptors can be used to assess their reactivity [5]. Secondly, and when the metabolic profile of a new compound is still unknown, its potential to yield reactive metabolites can be estimated by considering the occurrence of functional groups (the so-called structural alerts) which are known to generate RMs based on mechanistic studies on known drugs associated with idiosyncratic reactions or other toxicity profiles. In drug development strategies, these structural alerts should always be avoided regardless of the advantages they would offer [6]. Clearly, this second scenario is more frequent because toxicological screening is usually performed in the early phases of drug development when the number of drug candidates involved is too high to permit extended experimental metabolic studies [7].

While toxicity profiles (as expressed by LD_50_ values) are often predicted by correlative analyses using structural and physicochemical descriptors [8], such molecular properties have been rarely used to predict the ability of a given substrate to generate RMs, except for a few studies based on similarity descriptors [9]. This lack can have a double explanation. On the one hand, the various classes of structural alerts seem to be better related to the ability of a given compound to yield or not reactive metabolites. On the other hand, there may be doubts that a given compound should possess structural features able to forecast the involvement of toxication reactions.

The present study investigates the feasibility of predicting the ability of a given molecule to yield reactive metabolites by using physicochemical and stereo-electronic descriptors. The predictive models were generated by using a purposely developed classification algorithm based on enrichment factor optimization (EFO) and implemented in the VEGA suite of programs [10]. The study takes advantage from the already reported database which includes metabolic data as collected by manually curated analysis of the primary literature as published in the years 2004–2009. In detail, the database contains 1171 substrates (drugs and xenobiotics) which yield 6767 metabolic reactions and includes information about each included metabolite being (or not) a reactive molecule [11,12]. The present study is focused on the substrates giving reactive products in the first metabolic generation, namely in the metabolites directly deriving from parent compound (138 molecules out of 977, i.e., 14.1%) as well as in all generations (217 molecules out of 977, i.e., 22.2%). In detail, the first-generation RMs are primarily produced by the oxidation of unsaturated carbon atoms which represent the most abundant function (28%), followed by oxidations of nitrogen atoms (18%) and quinone formations (17%).

## 2. Results and Discussion

The predictions reported below involved both substrates yielding RMs in the first-generation as well as those giving RMs in any generation. Indeed, one may suppose that the properties of a given substrate might someway anticipate the reactivity of metabolites formed directly from it, while such properties should be less effective in predicting the reactivity of metabolites, which are indirectly generated in the subsequent generations. However, the predictions reported here involved also substrates giving RMs in any generation, by considering that an optimal model should be able to predict the capability of a given molecule to yield reactive metabolites regardless of the involved metabolic generation. Moreover, and focusing on first-generation RMs, specific models were also developed by separately considering the reactive metabolites produced by some specific metabolic reactions. 

As mentioned earlier, the predictions were repeated by simulating the substrates in their neutral and ionized forms. In detail, the used database includes a significant amount of ionizable molecules (315 out of 977 among which 215 basic compounds and 100 acid substrates); notably the ionizable molecules which yield RMs represent the 13.7% (43) a percentage truly superimposable to that seen in the whole database thus suggesting that the ionization characteristics do not influence the propensity to yield RMs. The relation between ionization state and predictive power should reveal whether molecular ionization can bias some relevant stereo-electronic properties (such as those derived by HOMO/LUMO energies) rendering them less efficient in properly accounting for chemical reactivity. Stated differently, if neutral molecules provide encouraging results, focusing only on neutral state might be advisable to avoid incorrect protonation states for molecules with complex ionization equilibria.

The study was organized in two parts: the first part involved a set of calibration analyses aimed at investigating the effect of the key parameters influencing the here reported classification approach as well as the role of ionization state in developing the predictive models. Based on these preliminary results, the second part will involve the generation of optimized predictive models that will be then compared with those which can be generated by applying a set of well-known classification algorithms as implemented in Weka 3.8 software [13].

Apart from for the models developed in the first part, which involved the entire dataset, the dataset was randomly subdivided into a learning and a test set (by default 70% and 30%). This was done by using a specially developed VEGA script (Training and test set creator.c), the models being generated using only the learning set, followed by their validation using them to predict the molecules yielding RMs included in the test set. To minimize the influence of randomness, this task was repeated 5 times. The results below describe the best predictors obtained by this approach.

### 2.1. Calibration (Preliminary) Analyses

Since these preliminary analyses have the primary objective to calibrate the algorithm parameters, the model generation was performed for simplicity on the entire dataset avoiding validation procedures. Among the user-defined parameters able to influence the proposed classification algorithm, attention was focused here on four key parameters, namely (1) the size of the cluster by which the quality function is calculated (see Equation (1)), (2) the exhaustiveness of random sampling, (3) the filtering cut-off in the top 5% enrichment factor below which a variable is discarded, thus influencing the number of considered descriptors, and (4) the number of variables included in each model. All these initial analyses were performed by predicting the substrates which gave RMs in their first generation, and by considering the compounds in their neutral state.

Table 1 shows the results of these calibration analyses obtained by monitoring the performance of the generated classifiers as parameterized by three relevant metrics: the average values for the 20 considered models of the substrates giving RMs in the top 1% and in the top 10% of the corresponding rankings, and the highest number of substrates found in the top 10%. These two percentages were chosen because the enrichment factor as computed in the top 1% encodes the ability of the method to concentrate “active” molecules in the top of the ranking, a feature which is particularly relevant in typical virtual screening campaigns, while the top 10% corresponds to the percentage of the ranking which is particularly relevant in these preliminary analyses since it roughly corresponds to the number of substrates giving first-generation RMs. Indeed, the dataset includes 138 RM-yielding substrates out of 978 compounds. These numbers imply that the RM-yielding compounds represent about 10% of the dataset.

The performances, as encoded by the mean percentage of substrates giving RMs in the top 1% and top 10% as well as highest number of “positive” substrates in the top 10%, are evaluated by exhaustively varying the cluster size, the sampling cycles, the number of included variables, the cut-off of the preliminary filtering to discard uninformative descriptors (in parenthesis, the number of discarded descriptors when lowering the threshold cut-off value) and the ionization state of the substrates. Notice that in these analyses the mean and best Top 10% correspond to the mean and best model sensitivity. N and I stand for substrates simulated in their neutral and ionized forms, respectively.

Regarding the effect of cluster size, Table 1 reveals that the two monitored enrichment factors (EFs) show contrasting trends: indeed, the EF in the top 1% improves when cluster size is reduced yet becomes worse when increasing the cluster size. In contrast, the EF in the top 10% parallels the cluster size, reaching a maximal value in clusters with a size equal to that of the considered top 10%. As discussed under methods, these preliminary analyses confirm that the cluster size must be almost equal to the number of positive compounds included in the used dataset. Based on these results, the following preliminary analyses were carried out by constantly considering the cluster size to be equal to 100.

The exhaustiveness of the random sampling is encoded by the number of sampling cycles performed to generate each model. This parameter is equal to 12 by default; but, as shown in Table 1, model generation involved doubling or halving the number of sampling cycles. The calculations seem to be modestly influenced by this parameter and show roughly constant results. However, it should be noted that a lower number of cycles speeds up calculations but unavoidably increases the randomness of the results, thus reducing their reproducibility. Hence, the proposed default value appears to be a reasonable compromise, which can be cautiously lowered when generating classifiers including either many variables or involving very extended dataset of descriptors to reduce the computational cost.

The role of the criterion by which the descriptors were filtered was investigated by progressively lowering the cut-off in the Top 5% enrichment factor below which a variable is discarded from 2.5 to 0.0 (default value = 2.0). As reported in Table 1, the number of discarded descriptors is proportional to the considered cut-off ranging from 18 (cut-off = 2.5) to 0. Nevertheless, increasing the number of considered descriptors does not enhance the performances of the models but induces a modest negative effect, which is more evident when lowering the cut-off below 1.5. More generally, the results show that filtering descriptors using a cut-off equal to 2.0 can extract the most informative variables and suggests that this cut-off could be slightly increased to speed up the calculations involving very large sets of descriptors.

Conceivably, the performances of the generated models increase with the number of included variables, even though Table 1 shows that marked statistical enhancements are seen up to 4 variables, while the inclusion of additional variables induces more limited improvements probably due to overfitting problems. Thus, classifiers including five or (at most) six variables should represents an optimal balance between computational time, predictive power, and robustness of the classifiers. In contrast, the generation of models with more variables requires a computational cost, which is not justified by the marginal increase of the corresponding performances as clearly witnessed by the models including 8 or 10 variables.

Table 1 shows that the influence of ionization state on model performances is clearly limited; ionized substrates afford slightly better results when considering classifiers with few variables, while more complex models show almost identical performances regardless of the substrate’s ionization state. When considering that the abundance of ionizable molecules within the dataset, the role of ionization state deserves further investigations and therefore the following models will be generated considering in parallel neutral and ionized substrates. Based on these preliminary analyses and to speed up the model generation, the following predictive analyses were carried out by considering: (a) a cluster size roughly equal to the number of “positive” substrates; (b) sampling cycles equal to 12; (c) filtering cut-off equal to 2.0; (d) classifiers including six variables. 

### 2.2. Predictive Models

#### 2.2.1. Classifiers for Substrates Yielding RMs in the First and Any Generation

As mentioned above and reported in Table 2, the second part of the study aimed at comparing optimized classifiers able to predict: (a) substrates giving first-generation RMs; (b) substrates giving RMs in any generation; substrates giving first-generation RMs through specific metabolic reactions such as (c) oxidations of Csp^2^ and Csp atoms, (d) oxidations to quinones or analogues and (e) oxidations of NH or NOH moieties.

Regarding the predictive models for substrates giving first and any generation RMs, Table 2 compiles the four best classifiers (Mods. 1–4) as obtained by (1) using the training set which is randomly collected and comprises the 70% of the entire dataset, (2) including six variables, (3) considering cluster size roughly equal to the number of active molecules in the training set (i.e., 70 or 140) and (4) simulating the substrates either in their neutral or ionized states. Regardless of the cluster size, the performances of a given classifier will be evaluated from the number of positive compounds falling in the first n position of the ranking (where n is the number of positive instances, Table 1). This number corresponds to the true positives; from it, the entire confusion matrix can be easily calculated since: (a) false positives = total positives − true positives; (b) false negatives = cluster size − true positives; (c) true negatives = total molecules − other three computed values. Table 2 reports the statistics of the selected classifiers as assessed by using them in predicting the molecules within the test set and reveals marked differences depending on both the ionization state and the involved generation. Indeed, while the statistics reported in Table 1 suggested that neutral and ionized substrates perform equally and can predict RMs generation with similar reliability regardless of the involved generation, Table 2 shows that (a) neutral substrates afford a better predictive power, and (b) the classifiers focused on the first-generation RMs perform better than those involving all generations.

In detail, the better performances of neutral substrates can be interpreted by considering that molecular charge affects some key stereo-electronic descriptors and hampers a precise evaluation of molecular reactivity. Not to mention that simulating neutral substrates represents a straightforward procedure which greatly simplify the calculations especially for molecules endowed with complex ionization equilibria. Again, Table 2 emphasizes that the molecular properties of a given substrate can account for the reactivity of the metabolites directly formed from it, while the RMs formation in the subsequent generations is conceivably less easily predictable. As a result, the best performing model (Mod. 1) will be used as a benchmark in extensive comparisons with the corresponding models as generated by using well-known classification algorithms implemented in the Weka software 

Regarding the included descriptors, all models feature a combination of physicochemical and stereo-electronic descriptors. There are four common variables which are included in at least 3 out of 4 models: (1) lipole, which is the only descriptor shared by all equations and which indirectly encodes for both molecular polarity and lipophilicity distribution [14]; (2) piS_total, which is the molecular self-polarizability and accounts for the reactivity of π electron systems as proposed by Coulson and Longuet-Higgins [15]; (3) De_total, which is the molecular electrophilic delocalizability according to Schüürmann [16], while (4) Electronic_energy is related to molecular reactivity. Moreover, all models include descriptors variously related to H-bonding capacity which might encode for both the polarity and the presence of easily oxidizable moieties. In detail, the generated classifiers suggest that the probability to give RMs increases with the reactivity of the aromatic moieties as well as with the molecular electrophilicity and apolarity. This results are understandably by considering that (1) all RMs are basically reactive electrophilic compounds and thus substrates which already possess a marked reactivity (parameter #4) and electrophilicity (parameter #3) will be more prone to generate RMs; (2) many RMs arise from the oxidation of aromatic rings to quinones and analogues thus justifying the occurrence of a descriptor specifically related to the reactivity of π electron systems (parameter #2) (3) parameters encoding apolarity and H-bonding capacity can be seen as a measure of the propensity of a given molecule to undergo oxidative metabolic reactions (by which RMs are produced).

#### 2.2.2. Classifiers for Specific Metabolic Reactions

Preliminary models were developed with a view to identifying the first-generation RMs generated by specific metabolic reactions from among all considered substrates, but these initial analyses proved unsuccessful (models not shown). Along with the above-mentioned problem of the unbalanced datasets which here appears to be particularly exacerbated, such a failure can be explained by considering that such models in fact involve two distinct predictions, namely which substrates yield RMs and which substrates undergo a specific metabolic reaction. Reasonably, these two distinct features can depend on different (and maybe contrasting) molecular properties thus justifying the unsatisfactory results. 

More homogeneous models should predict which substrates form RMs through a specific metabolic reaction either from among all substrates undergoing the same specific reaction or from among all substrates generating first-generation RMs. The first type of prediction still involves the recognition of substrates yielding RMs, and thus resembles those already developed in the previous sections even though focused on a specific subset of all simulated molecules. In contrast, the second type of prediction appears conceptually different compared to the previous ones, since it predicts the susceptibility of a given molecule to undergo a specific metabolic reaction. Hence, the following analyses will be focused on the second prediction type both for its novelty and because its results can be combined with the previous models offering a kind of predictive procedure by which one may first predict which molecules can yield RMs and then through which reaction(s) they can be generated. 

Moreover, the second prediction type has the added benefit of involving clearly less unbalanced datasets since here the positive compounds are 39, 23 and 24 out of the 138 first-generation RMs as produced by oxidation reactions of Csp^2^ and Csp atoms, to quinones or analogues, and of NH or NOH moieties, respectively. Based on the previous results and focusing on the randomly generated training set, the cluster size is equal to 25 for the first reaction type (Mods. 5 and 8, Table 2) and 20 for the other two cases (Mods. 6, 7, 9 and 10, Table 2), while the initial filtering of the variables based on their enrichment factor on the Top 5% was rendered less stringent (the required EF value equal to 1.0 instead of 2.0) to avoid an excessive reduction in the number of descriptors considered.

Table 2 compiles the best classifiers as generated by considering either neutral or ionized substrates. As a trend, the obtained models show truly satisfactory statistics as emphasized by the corresponding MCC values always greater than 0.5. Conceivably, these remarkable results benefit from using markedly smaller and less unbalanced learning sets compared to the previously used datasets and these results suggest also the here proposed approach is influenced by the composition of the learning sets even though additional tests involving very unbalanced datasets (as used, for example, in virtual screening campaigns) should be required to precisely assess the performances and limitations of the EFO method. More importantly, these notable models bear witness to the possibility of successfully predicting the specific metabolic reaction(s) a given substrate may undergo, considering only physicochemical and stereo-electronic descriptors. We note that such type of prediction could find more general applications in predicting the metabolism of xenobiotics. 

In more detail, the models compiled in Table 2 allow for some interesting observations. Apart from the models obtained to recognize the substrates undergoing oxidative reactions at Csp^2^ and Csp atoms, the ionized substrates perform better than the neutral ones. This finding is in contrast to the results so far reported and can be explained by considering that here the best performing ionization state should correspond to that concretely involved in molecular recognition by the relevant metabolizing enzymes, while the previous predictions mostly depend on the intrinsic reactivity of a molecule that is less influenced by the simulated ionization state as evidenced in the previous sections. On these bases, the obtained results suggest that the ionized forms play a role in quinone formation and more markedly in NH/NOH oxidation where the ionization equilibria can directly affect the sites of metabolic attack (compare Mods. 7 and 10).

Almost all obtained models include a proper combination of physicochemical and stereo-electronic parameters even though the relevance of the latter is here clearly less pronounced than in the previous classifiers, a result particularly evident for the descriptors featuring the HOMO/LUMO energies. This finding can be explained by considering that HOMO/LUMO energies and their derived parameters are particularly informative in predicting chemical reactivity, while here physicochemical properties are more convenient in describing the recognition between substrates and enzyme. Clearly, some stereo-electronic parameters are also included in these last models where they reasonably account for the covalent phases of the enzymatic reactions as seen for electrophilicity indices when predicting substrates undergoing NH/NOH oxidation (Mod. 8), as well as the LUMO energies for substrates undergoing Csp^2^/Csp oxidation (Mod. 7).

#### 2.2.3. Comparison of the Best Model with Weka Results

With a view to further evaluating the performances of the here proposed EFO classification method, the best performing model (namely the prediction of neutral substrates yielding first-generation RMs, Mod. 1) was compared with the corresponding models as generated by using 29 different classification algorithms implemented in the Weka suite of programs. Table 3 reports the corresponding MCC values and clearly reveals that most tested algorithms provide models with a predictive power significantly lower than Mod. 1 and only 6 out of 29 compared models show an MCC value ≥ 0.30.

Notably, linear discriminant analysis (LDA) and the functional linear discriminant analysis (FLDA) [17], which are very popular approaches to predict categorical features. using continuous variables, perform markedly worse than the here proposed algorithm even though the model obtained with FLDA shows an MCC (2.6) close to the mentioned threshold of 0.3. Among the algorithms which surpass this threshold, the k-NN classifiers (IBk) [18] and the randomizable classifiers [19] generate models with performances very similar to those of Mod. 1 in terms of both the MCC value and the number of true positives. Finally, tree algorithms offer the best performances among the methods included in the Weka software: in detail, the classifier based on the pruned J48 algorithm [20] shows a comparable MCC value and a higher number of true positives compared to Mod. 1, while the Random Forest method [21] yields the highest MCC value but a lower number of true positives.

Taken together and although the compared Weka models were developed by adopting the included default parameters (namely without optimization procedures), the comparison described above reveals that the proposed method shows performances comparable to, or only slightly worse than, the best available classification algorithms. Moreover, it should be noted that the tree algorithms generated slightly better models by including all available descriptors, while Mod. 1 involves only six descriptors, a difference that should avoid overfitting issues rendering Mod. 1 more robust and more extensively applicable. Again, the proposed approach affords a score-based instead of a simple binary prediction and this means that the better the score, the higher the probability that a given substrate generates RMs.

For example, in the above reported analyses, the compounds classified in the top 10% are considered as yielding RMs, and Mod. 1 places 48 true positives in this best cluster. This means that Mod. 1 shows a sensitivity of 0.48, meaning that if a molecule falls in the top 10% it has a probability of 48% to yield one or more RMs. Nevertheless, this probability is not constant but depends on the computed score and indeed if a given molecule has a score that brings it in the top 2% the probability of giving RMs raises to 63%. Similarly, if a substrate falls in the remaining 90% it is predicted as non-reactive and Mod. 1 has a specificity equal to 0.9, meaning that this predicted probability is equal to 90%. However, if a given molecule falls in the bottom 10% this probability increases to 97%. In other words, the proposed method allows a score-based probability to be assigned to each prediction. This can represent a crucial advantage compared to most available classification approaches.

### 2.3. General Applicability in Machine Learning Analyses

With these encouraging results in hand, the last part of the study applied the EFO approach to UCI datasets which are routinely used to test new machine learning algorithms as collected in http://archive.ics.uci.edu/ml/index.php. These analyses had two primary objectives since they were planned to test the EFO performances when using (1) non-cheminformatics data and, more interestingly, (2) roughly balanced datasets. Among the available datasets, attention was focused on two balanced datasets chosen because they were recently used for benchmarking analyses in a study to validate a new method for generating training and test sets and thus the here obtained results can be easily compared to the published models [22]. Moreover, the two chosen datasets involve categorical predictions based on categorical, integer and real attributes. In detail, the first dataset comprises various health data for heart patients and the predicted attribute refers to the occurrence of heart disease in the collected patients [23]. The second dataset involves sonar signals and the predicted attribute is the discrimination between metals or rocks based on a pattern of 60 frequency-modulated signals in the range 0.0 to 1.0 [24]. 

As already observed when predicting specific metabolic reactions, the preliminary filter based on the EF value of each attribute should be carefully tuned when analyzing less unbalanced dataset based on the calculation of the maximum EF reachable. For example, the maximum EF value for a perfectly balanced dataset is equal to 2.0 and this suggests that the above defined default threshold value is unsuitable, and the filter should be either removed or markedly smoothed. In detail, the predictions of these last datasets were performed adopting cut-off values equal to 1.5, 1.0 and 0.0 (i.e., no filter).

Table 4 reports the major characteristics of the used datasets and compares the performances (in terms of Accuracy values) of three well-known classification algorithms (i.e., C4.5, NB, k-NN as taken from) with those obtained using the EFO approach. The obtained results emphasize that the EFO method compares with the other approaches and afford encouraging results for both datasets. The tested threshold values suggest that the initial filter can be conveniently removed when analyzing balanced datasets including a reduced number of variables (as done for the heart dataset). Nevertheless, less stringent filters can still be suitable when the high number of considered variables could excessively slow down the calculations (as done for the sonar dataset).

Remarkably, the EFO method affords truly interesting results for the heart dataset performing slightly better than the other three algorithms. An extended comparison of the here obtained statistics with those reported in literature revealed that very few algorithms can provide better models (e.g., see ref. [25]). For example, a very recent study reported that the approaches based on neural network can generate models with accuracy values > 0.9 [26].

Table 2 reports the best performing model (Mod. 11). Apart from the maximum heart rate achieved, all included variables increase the probability of heart disease. Specifically, and in the used test set, 36 out of 42 instances with a Score < −4.0 and all 25 instances with Score < −12.0 were heart patients thus suggesting that these values can represent easily computable thresholds by which the occurrence of heart diseases can be successfully predicted.

## 3. Methods

### 3.1. Dataset Set-Up

As mentioned in the Introduction, the study involved 977 compounds, the 3D structure of which was either generated manually using the VEGA software or, when available, automatically retrieved from PubChem. The molecules were simulated both in their neutral form and in their preferred ionization state as existing at physiological pH. Their conformation and atomic charges were optimized and refined by PM7 semi-empirical methods as implemented in MOPAC 2016 [27] which also allowed the calculation of a relevant set of stereo-electronic descriptors including, among others, the HOMO/LUMO-based reactivity indices and the delocalizability descriptors. The minimized conformations were then used by VEGA to calculate an extended set of geometrical and physicochemical descriptors by discarding highly correlated variables. In this way, a set of 28 descriptors was collected and used in the development of the predictive models as described below. The computed descriptors were directly used in the study without scaling, weighting, or normalization procedures. The dataset used in the predictive analyses for neutral substrates is collected in Appendix A.

### 3.2. Classification Algorithm

Given the well-known limitations of the common classification algorithms in providing satisfactory results when, as in this study, the learning set is markedly unbalanced, a classifier method based on logistic regressions as driven by an enrichment factor optimization (EFO) has been developed and included in VEGA ZZ package as the Automatic model builder.c script. Such a method predicts a categorical dependent variable (the RMs generation) by developing linear combinations of continuous independent variables (the molecular descriptors). Since the developed equations produce continuous output values and not the expected binary outputs, during the learning phase the *n* best score compounds are considered as positives and the remaining (*t* − *n*) compounds are considered as negatives (where *n* is the number of positive compounds included in the training set and *t* is the total number of instances). For example, the prediction of substrates generating first-generation RMs assumed the 138 compounds with the best scores as yielding RMs, while the remaining 839 molecules are considered as non-reactive substrates. The resulting classifiers were thus evaluated by considering their capacity to place the RM-yielding molecules within the 138 top positions in the ranking. Accordingly, the score computed for the 138th compound, namely the last compound considered as positive, represents a threshold value, which will allow the discrimination between positives and negatives in the validation phase and, more in general, when applying the obtained model to external compounds.

As schematized in Figure 1, the algorithm proposed here is composed of several logical units starting from a preliminary data filtering which allows the selection of the most informative descriptors. In such an initial process, each descriptor is filtered based on its capability to place the RM-yielding substrates in the top of the ranking by simple EF analysis. Only descriptors with an EF value as computed for the top 5% greater than a user-defined threshold (by default equal to 2.0) were selected for the model generation.

Hence, the selected descriptors were systematically combined to generate classification models according to the user-defined number of independent variables. The coefficients of the resulting equations are calculated by applying the Hooke-Jeeves optimizer for non-continuous functions, the goal being to optimize the ranking position of the substrates yielding RMs. Moreover, a random sampling algorithm is applied to evade local minima thus better optimizing the resulting classifiers. The performances of each resulting model are evaluated by a purposely defined quality function.

Similarly to virtual screening metrics, the ability of a classifier to correctly recognize the relevant compounds can be described by two kinds of parameters: firstly, enrichment factors account for the capacity to focus the correct compounds on the top of the ranking without considering what happens in the remaining part of the ranking; secondly and in contrast, the metrics variously based on receiver operating characteristic (ROC) curves evaluate the reliability of the entire ranking but fail to parameterize how many substrates are correctly classified in the first (best) positions (the so-called early recognition problem).

As proposed in a previous study [28], cluster analysis can afford a graphically intuitive way to evaluate the overall reliability of a classifier by monitoring how many RM-yielding compounds are included in each cluster. A suitable model should be able to place most of the RM-yielding substrates in the first cluster, with their proportion progressively decreasing in the following clusters. In contrast, models which randomly distribute the RM-yielding substrates in all clusters should be considered unsatisfactory regardless of how many correct molecules are placed in the best clusters.

Besides offering a graphical evaluation, the distribution of substrates which yield reactive metabolites can be used to derive a quantitative parameter based on the asymmetry index (AI), a measure of deviation of the cluster distribution from a normal curve which can be computed based on the Pearson’s moment coefficient of skewness [29]. Thus, the greater the AI value (namely the more right-skewed the distribution) the better the model. Moreover, and to also optimize the early recognition, the quality function optimized in the model generation is calculated as the product of the asymmetry index and the percentage of RM-yielding substrates included in the first (best) cluster (RM_1_). This is reported in Equation (1) where *n* is the number of clusters, RM_i_ is the abundance of substrates yielding RM in the *i* cluster and RM_m_ is the RM average.
(1)Quality=∑i=1n1n(RMi−RMm)3(∑i=1n1n(RMi−RMm)2)32 × RM1 

In the following analyses, the size of the cluster will be defined as roughly equal to the number of RM-yielding substrates so that RM_1_ encodes the capacity of the model to discriminate between substrates yielding or not reactive metabolites. Although maximizing the distance between “active” and “inactive” instances is the primary objective of the classification algorithm, the quality function also includes the asymmetry index to assure that the frequency of RM-yielding substrates decreases when moving towards the bottom of the ranking. In this way, the entire resulting ranking encode for a sort of probability score, which can be associated with each binary prediction and can become particularly insightful for the doubt cases. 

The graphical interface of this tool allows the selection of: (i) input and output files; (ii) the dependent variable (here the Boolean RM value) with a cut-off above which a RM-yielding compound should be considered as active (in the case of Boolean properties, a cut-off value equal to 0.5 can be set); (iii) the independent variables which can be selected from the specified input file; (iv) the number of variables to be included in the model; (v) the size of the clusters into which the ranking should be subdivided when calculating the quality function. Moreover, the user can define the number of cycles of random sampling performed to generate each starting model (by default 12 for each included variable) as well as the characteristics of the optimization algorithm (iterations = 5000, RMS = 0.001).

At the end of calculations, the resulting output files comprise: (i) a file containing the results for the selected best models, (ii) a log file including the details of the performed calculation, (iii) a file compiling the computed scores for each molecule and for each model and (iv) a reduced input file including only the best performing descriptors. This last file can be used as input to speed up the calculations in which models including several variables are generated and for which an exhaustive model generation could become too time-consuming. The pseudo-code of the entire algorithm is included in the Appendix A.

## 4. Conclusions

The study proposes a novel classification algorithm based on linear combinations of descriptors, which are generated through enrichment factor optimization (EFO). Even though this approach could find many insightful applications in virtual screening campaigns, it is here presented by considering its potential as a general classification approach in predicting substrates yielding RMs. The study takes advantage from a previously collected and reported metabolic database and reveals that even though most hitherto published predictive models are based on the occurrence of well-defined structural alerts, the capacity of a given substrates to form RMs (at least in the first-generation metabolism) can also be predicted by using physicochemical and stereo-electronic descriptors with the latter playing a key role in parameterizing the intrinsic reactivity of a molecule and its metabolites. As an aside, the study also comprises classifiers able to recognize the kind of metabolic reaction a molecule can undergo, and these preliminary results open the door to the use of this approach in metabolism predictions. More generally, the predictive models developed here emphasize the potential of using highly curated metabolic datasets and suggest that the exploited database can provide reliable learning sets for developing various metabolic predictive models. Moreover, and for simplicity, the here proposed models were developed focusing on the lowest energy conformation even though one may argue that monitoring more than one representative geometry might improve the models especially for very flexible molecules.

The comparison of the generated best performing model with the classifiers developed by using different classification algorithms implemented in the Weka software reveals that the proposed approach compares with the best available approaches and shows two crucial advantages since it involves a limited number of descriptors and provides a score-based probability which allows a critical evaluation of the obtained prediction.

Finally, the last analyses focused on non-cheminformatics data emphasize the general applicability of the EFO method which provide satisfactory results even when using balanced datasets regardless of the type of included variables. Specifically, the EFO application on the heart dataset afforded a highly performing model, which allows a very easy and successful prediction of the occurrence of heart disease.

## Figures and Tables

**Figure 1 molecules-23-02955-f001:**
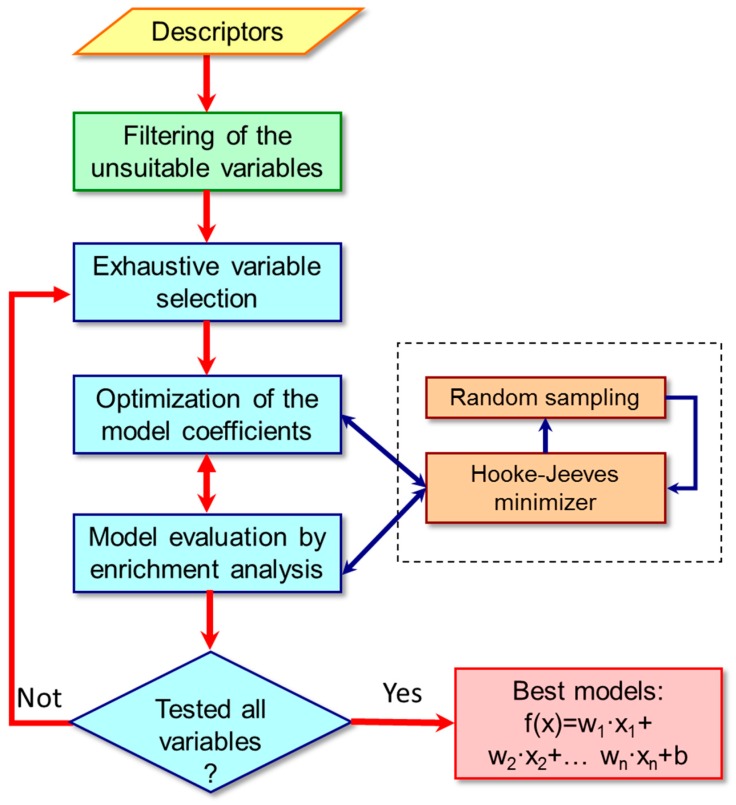
Main logical units into which the proposed classification algorithm can be subdivided. The yellow box indicates the input, the green box comprises the initial variable filtering; the blue boxes define the main tasks performed by the algorithm; the red box displays the obtained results. The brown boxes include the computational approaches by which each generated classifier is optimized by maximizing the corresponding quality function.

**Table 1 molecules-23-02955-t001:** Model performances obtained in the calibration study.

Cluster Size	Sampling Cycles	Variables	EF Cut-Off Top 5%	Ionization State	Mean Top 1%	Mean Top 10%	Best Top 10%
10	12	3	2.0	N	82.2	25.95	32
20	12	3	2.0	N	68.9	28.2	34
40	12	3	2.0	N	59.4	32.7	42
60	12	3	2.0	N	58.3	34.5	45
80	12	3	2.0	N	55.7	36.1	46
100	12	3	2.0 (12)	N	47.8	38.2	46
100	6	3	2.0	N	48.8	38.2	47
100	24	3	2.0	N	47.2	37.9	46
100	12	3	2.5 (18)	N	44.4	38.1	45
100	12	3	1.5 (6)	N	48.8	36.4	45
100	12	3	1.0 (2)	N	49.4	32.5	43
100	12	3	0.0 (0)	N	51.1	32.7	43
100	12	1	2.0	N	37.4	25.1	30
100	12	2	2.0	N	49.4	32.2	40
100	12	4	2.0	N	49.9	39.7	45
100	12	5	2.0	N	51.1	41.6	47
100	12	6	2.0	N	55.5	43.6	48
100	12	8	2.0	N	56.7	44.3	48
100	12	10	2.0	N	57.2	45.5	48
100	12	2	2.0	I	44.4	34.3	42
100	12	3	2.0	I	47.2	39.4	46
100	12	4	2.0	I	49.4	40.6	47
100	12	5	2.0	I	50.2	42.2	47
100	12	6	2.0	I	54.4	44.0	48

**Table 2 molecules-23-02955-t002:** Best developed models and relative statistics.

Mod.	Gen./React.	State	Cluster Size	Equation	Statistics
1	First	N	70	1.00 HBT + 2.47 Lipole − 0.0001 Electronic_Energy + 0.13 Dipole + + 2.55 Dn_Total − 2.73 De_Total	Precision = 0.42 Accuracy = 0.85 MCC = 0.33
2	First	I	70	1.00 Rotors − 1.55 HBA + 5.09 Lipole − 0.0018 Electronic_Energy + +5.40 Dn_Total − 5.22 PiS_Total	Precision = 0.35 Accuracy = 0.81 MCC = 0.24
3	All	N	140	1.00 HBA + 1.09 Lipole − 0.0089 Heat_Formation + 0.070 Filled_Levels + − 2.03 De_Total + 4.47 PiS_Total	Precision = 0.42 Accuracy = 0.77 MCC = 0.28
4	All	I	140	1.00 Lipole − 0.033 PSA − 0.0059 ASA − 0.0004 Electronic_Energy + − 0.23 De_Total + 2.04 PiS_Total	Precision = 0.46 Accuracy = 0.74 MCC = 0.29
5	Csp^2^/Csp ox	N	30	−1.00 Angles + 19.13 Rotors − 0.43 HBA + 15.47 HBT − 9.89 Impropers + + 21.32 Lipole	Precision = 0.67 Accuracy = 0.83 MCC = 0.55
6	Quinone ox	N	20	1.00 Angles + 1.07 Rotors + 68.34 Radius_Gyration − 8.38 HBA + − 30.28 HBD − 1.09 ASA	Precision = 0.63 Accuracy = 0.87 MCC = 0.54
7	NH/NOH ox	N	20	−1.00 HBD + 0.041 Impropers − 0.15 Dipole − 0.0007 E_HOMO + + 0.68 Mulliken_Electronegativity + − 0.46 Schuurmann_alpha	Precision = 0.63 Accuracy = 0.87 MCC = 0.54
8	Csp^2^/Csp Ox	I	30	−1.00 Rotors − 10.12 HBA + 1.47 HBD + − 1.36 Impropers + 1.04 PSA + + 0.40 E_LUMO	Precision = 0.61 Accuracy = 0.78 MCC = 0.46
9	Quinone ox	I	20	1.00 Angles + 1.53 Rotors + + 14.46 Radius_Gyration − 0.42 HBA + − 14.65 HBD − 0.46 ASA	Precision = 0.67 Accuracy = 0.83 MCC = 0.55
10	NH/NOH ox	I	20	−1.00 HBD + 0.0083 Impropers + − 0.20 Lipole + 0.035 LogP_MLP_ + + 0.015 Dipole + 0.14 Ionization_Potential	Precision = 0.70 Accuracy = 0.87 MCC = 0.61
11	Heart data	N/A	75	−1.00 Pain + + 0.063 maximum_heart_rate_achieved + − 0.47 exercise_induced_angina + − 2.07 oldpeak + − 2.68 number_of_major_vessels + − 1.54 thal	Precision = 0.86 Accuracy = 0.87 MCC = 0.71

N and I stand for substrates simulated in their neutral and ionized forms, respectively.

**Table 3 molecules-23-02955-t003:** Comparison of the predictive power (as encoded by MCC value) of Mod. 1 with the corresponding models obtained by using 29 different algorithms as implemented in Weka software.

Algorithm	MCC	Algorithm	MCC
**Mod. 1**	**0.33 (21)**	IterativeClass	0.24
BayesNet	0.12	RandomSubspace	0.16
FLDA	0.25	DecisionTable	0.13
LDA	0.17	JRip	0.14
Logistic	0.11	PART	0.19
**Multilayer**	**0.30 (7)**	DecisionStump	0.13
**IBk**	**0.33 (14)**	**J48**	**0.33 (17)**
Kstar	0.27	LMT	0.17
LWL	0.13	**RandomForest**	**0.37 (9)**
AdaBoostM1	0.13	RandomTree	0.16
Bagging	0.21	REPTree	0.14
Regression	0.27	LogitBoost	0.18
FilteredClass	0.16	**Randomcommitee**	**0.32 (13)**
A1DE	0.14	**NNge**	**0.30 (7)**
CHIRP	0.23	ExtraTree	0.25

The methods affording an MCC value ≥ 0.30 are indicated in bold and for these best performing approaches, the obtained number of true positives in the test set is reported between parentheses. Notice that the approaches providing models with MCC < 0.1 are not reported for simplicity.

**Table 4 molecules-23-02955-t004:** Comparison of the here obtained performances (in terms of accuracy) with those published in ref. 28 (i.e., C4.5, NB and k-NN) for the two used UCI datasets.

Dataset	Attributes	Instances	Accuracy					
			C4.5	NB	K-NN	EFO (0.0)	EFO (1.0)	EFO (1.5)
Sonar	60	208	0.68	0.71	0.84	-	0.76	0.69
Heart	13	270	0.74	0.86	0.59	0.87	0.73	0.73

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
