# Peer review of "Prediction of the Formation of Reactive Metabolites by A Novel Classifier Approach Based on Enrichment Factor Optimization (EFO) as Implemented in the VEGA Program"

_molecules, 2018, doi:10.3390/molecules23112955_

Round 1

Reviewer 1 Report

This paper reports the development of a number of linear classifier models on the basis of classification of properties using enrichment factor optimization (EFO) for the prediction of the formation of reactive metabolites in small molecule drugs. The approach described is interesting and is shown to be as successful as other existing methods. The work described is presented to a high level of standard.

It is common with this type of approach to end up with a list of models that contains various physicochemical and stereo-electronic molecular properties. While the authors have summarised that some of them are commonly found in most models, they should briefly explain why it makes sense for these properties (i.e. lipole, molecular self-polarizability, molecular electrophilic delocalizability and electronic energy), as opposed to others, to appear most consistently in the models.

The authors should also present representative examples of cases where their classifier models have consistently failed, and discuss why the models fail. This is important as it allows rationalization of why and when these models can fail.

Author Response

Response to Reviewer 2 Comments

This paper reports the development of a number of linear classifier models on the basis of classification of properties using enrichment factor optimization (EFO) for the prediction of the formation of reactive metabolites in small molecule drugs. The approach described is interesting and is shown to be as successful as other existing methods. The work described is presented to a high level of standard.

It is common with this type of approach to end up with a list of models that contains various physicochemical and stereo-electronic molecular properties. While the authors have summarised that some of them are commonly found in most models, they should briefly explain why it makes sense for these properties (i.e. lipole, molecular self-polarizability, molecular electrophilic delocalizability and electronic energy), as opposed to others, to appear most consistently in the models.

A sentence delving into the meaning of the descriptors included in the best models was added at page 6.

The authors should also present representative examples of cases where their classifier models have consistently failed, and discuss why the models fail. This is important as it allows rationalization of why and when these models can fail.

The difficulties found when predicting the specific reactions by considering the whole database were discussed by considering how also the here presented method is influenced by the database composition and might suffer with very unbalanced datasets. 

Reviewer 2 Report

The work of Mazzolari et al. by title “Prediction of the formation of reactive metabolites by a novel classifier approach based on enrichment factor optimization (EFO) as implemented in the VEGA program” reports the development of a new classification algorithm that is used to predict the generation of reactive metabolites for a given substrate. The algorithm is based on the use of the enrichment factor optimization (EFO) method and predictive models of metabolism are generated using a manually curated database containing 977 annotated substrates that have been collected from literature in 2004-2009. The performance of the algorithm is also tested with non-cheminformatics UCI datasets.

Overall, the work is well conceived and the article is clearly written. Results are compelling and show a good performance of the EFO approach. Nevertheless, the authors report some controversial result for metabolites simulated in their neutral and ionized forms, that needs some minor revision. Specifically, results of calibration study (Table 1) suggest limited influence of ionization state on model performances (page 5, line 178). Accordingly, the authors state that “simulating the substrates in their neutral form is an acceptable approximation that does not worsen the predictive models and can represent an efficient procedure”. In predictive analyses, however, significant differences are found between using neutral substrates and ionized substrates (Table 2), with the former affording better predictions. Authors then provide a tentative explanation for this observation, stating that “the better performances of neutral substrates can be interpreted by considering that molecular charge affects some key stereo-electronic descriptors and hampers a precise evaluation of molecular reactivity” (page 6, lines 214-216). When considering specific reactions (e.g. Quinone ox; NH/NOH ox), however, this seems not the case, as also stated by the authors at page 8, lines 270-279. In view of the results from the analysis of specific metabolic reactions and calibration study, the authors should revise the aforementioned statement at page 6, lines 214-216. Furthermore, since ionizable substrates represent 32% of the dataset, the authors should provide a more thorough discussion on the distribution of their substrates along calculated (or experimental) pKa values, analyzing the performance of the classifier for substrates yielding RMs and for specific metabolic reactions when considering acidic and basic compounds, as well as molecules with multiple pKa (i.e. substrate with complex ionization equilibria) as distinct groups of substrates.  How many acidic substrates, basic substrates, and multiple ionizable substrates are true positive in the confusion matrix? Does negative or positive molecular charge diversely affect the evaluation accuracy of molecular reactivity?

Author Response

Response to Reviewer 1 Comments

The work of Mazzolari et al. by title “Prediction of the formation of reactive metabolites by a novel classifier approach based on enrichment factor optimization (EFO) as implemented in the VEGA program” reports the development of a new classification algorithm that is used to predict the generation of reactive metabolites for a given substrate. The algorithm is based on the use of the enrichment factor optimization (EFO) method and predictive models of metabolism are generated using a manually curated database containing 977 annotated substrates that have been collected from literature in 2004-2009. The performance of the algorithm is also tested with non-cheminformatics UCI datasets.

Overall, the work is well conceived and the article is clearly written. Results are compelling and show a good performance of the EFO approach. Nevertheless, the authors report some controversial result for metabolites simulated in their neutral and ionized forms, that needs some minor revision. Specifically, results of calibration study (Table 1) suggest limited influence of ionization state on model performances (page 5, line 178). Accordingly, the authors state that “simulating the substrates in their neutral form is an acceptable approximation that does not worsen the predictive models and can represent an efficient procedure”. In predictive analyses, however, significant differences are found between using neutral substrates and ionized substrates (Table 2), with the former affording better predictions. Authors then provide a tentative explanation for this observation, stating that “the better performances of neutral substrates can be interpreted by considering that molecular charge affects some key stereo-electronic descriptors and hampers a precise evaluation of molecular reactivity” (page 6, lines 214-216). When considering specific reactions (e.g. Quinone ox; NH/NOH ox), however, this seems not the case, as also stated by the authors at page 8, lines 270-279. In view of the results from the analysis of specific metabolic reactions and calibration study, the authors should revise the aforementioned statement at page 6, lines 214-216. Furthermore, since ionizable substrates represent 32% of the dataset, the authors should provide a more thorough discussion on the distribution of their substrates along calculated (or experimental) pKa values, analyzing the performance of the classifier for substrates yielding RMs and for specific metabolic reactions when considering acidic and basic compounds, as well as molecules with multiple pKa (i.e. substrate with complex ionization equilibria) as distinct groups of substrates.  How many acidic substrates, basic substrates, and multiple ionizable substrates are true positive in the confusion matrix? Does negative or positive molecular charge diversely affect the evaluation accuracy of molecular reactivity?

The controversial sentences were modified as suggested by the reviewer and a sentence detailing the characteristics of the included ionizable molecules was added at page 3.